# Gene Expression in the Developing Seed of Wild and Domesticated Rice

**DOI:** 10.3390/ijms232113351

**Published:** 2022-11-01

**Authors:** Sharmin Hasan, Agnelo Furtado, Robert Henry

**Affiliations:** 1Queensland Alliance for Agriculture and Food Innovation, University of Queensland, Brisbane 4072, Australia; 2Department of Botany, Jagannath University, Dhaka 1100, Bangladesh; 3ARC Centre of Excellence for Plant Success in Nature and Agriculture, St Lucia 4072, Australia

**Keywords:** gene expression, RNA-Seq, seed development, differentially expressed genes, domestication loci, starch and sucrose metabolism, seed storage proteins, wild rice

## Abstract

The composition and nutritional properties of rice are the product of the expression of genes in the developing seed. RNA-Seq was used to investigate the level of gene expression at different stages of seed development in domesticated rice (*Oryza sativa* ssp. *japonica* var. Nipponbare) and two Australian wild taxa from the primary gene pool of rice (*Oryza meridionalis* and *Oryza rufipogon* type taxa). Transcriptome profiling of all coding sequences in the genome revealed that genes were significantly differentially expressed at different stages of seed development in both wild and domesticated rice. Differentially expressed genes were associated with metabolism, transcriptional regulation, nucleic acid processing, and signal transduction with the highest number of being linked to protein synthesis and starch/sucrose metabolism. The level of gene expression associated with domestication traits, starch and sucrose metabolism, and seed storage proteins were highest at the early stage (5 days post anthesis (DPA)) to the middle stage (15 DPA) and declined late in seed development in both wild and domesticated rice. However, in contrast, black hull colour (*Bh4*) gene was significantly expressed throughout seed development. A substantial number of novel transcripts (38) corresponding to domestication genes, starch and sucrose metabolism, and seed storage proteins were identified. The patterns of gene expression revealed in this study define the timing of metabolic processes associated with seed development and may be used to explain differences in rice grain quality and nutritional value.

## 1. Introduction

Rice (*Oryza sativa* L.) is a grass with a small genome (~430 Mbp) [1] that is a model for other major cereal crops (e.g., barley, wheat, maize, and sorghum) in terms of sequences, structure, order, and functions of genes. The seed is the progenitor of the new generation and contains a sizable source of nutrition required for both humans and animals [2]. Seed development is a crucial process in the lifespan of angiosperms. The rice grain is made up of three genetically different tissue types: the filial embryo, the triploid endosperm, and the maternal pericarp and seed coat [3]. Development of the embryo and endosperm happens in an organized and chronological manner following double fertilization [2]. After fertilization, many free endosperm nuclei are placed in the peripheral area of the embryo sac. Cell wall formation and mitotic stages begin at 3 days after fertilization (DAF) and 4 DAF, respectively. Endosperm cellularization starts in the second stage of seed development and completes at 6 DAF. While the initiation of embryo morphogenesis and endoreduplication, and the formation of milky endosperm occur in the third stage at 8–10 DAF [2]. Then, the transition of milky endosperm from soft dough to hard dough takes place, and as the maturation progresses, it becomes dry and dormant [4] with programmed cell death in the endosperm at 16 DAF [2].

During seed development, carbohydrates, amino acids, seed storage proteins, fatty acids, and other metabolites are accumulated in the developing seeds via different pathways. Most of the carbohydrate in the rice grain is starch, representing about 90% of the dry grain weight [5,6]. Starch is the main source of nutrition for seedling growth. Seeds contain seed storage proteins, including acidic or basic solution-soluble glutelins, alcohol-soluble prolamins, water-soluble albumins, and salt soluble α-globulins that provide nutrients to the seedlings for growth and development [7]. The glutelins make up 60–80% of the total endosperm protein and therefore, has been investigated most extensively [7]. The developmental changes in seeds are controlled by complex regulatory mechanisms. Functional understanding of the genes and knowledge of the patterns of gene expression associated with the stages of seed development is crucial in better defining these molecular and biochemical events [8] and may provide insight into how to improve the nutritional quality and yield of grains.

Transcriptome profiling has paved the way for transcript discovery and the quantification of gene expression [9]. In the early days, mRNA was quantified by northern blotting to measure the expression of specific RNAs [10]. Then, polymerase chain reaction aided transcript titration assay (PATTY) was developed to amplify distinct nucleic acid sequences to assess the expression of RNAs [11]. Sequencing-based gene expression analysis based on expressed sequences tags (ESTs) which were sequences of partial cDNAs produced from transcribed RNAs was applied [12]. EST-based methods including serial analysis of gene expression (SAGE) [13], and cap analysis of gene expression (CAGE) [14] were subsequently developed to rapidly detect expressed genes and gene isoforms. The quantitative real-time PCR (qRT-PCR) was used for gene expression analysis due to its high sensitivity and accuracy, good reproducibility, and rapid processing [15]. DNA microarrays (or “chips”) was developed to detect many genes simultaneously and became the most widely used method in transcriptomics [16]. Recent advancement in genomic tools and sequencing technology has facilitated comprehensive transcriptional characterization robustly and effectively. High-throughput RNA-sequencing (RNA-Seq) enables the provision of an inclusive picture of the transcriptome, allowing the complete annotation of the genes, novel genes, and alternatively spliced isoforms [17,18]. Increasing studies using transcriptome analysis based on the Illumina RNA-Seq platforms have shown that RNA-Seq is more appropriate and inexpensive for gene expression studies. This technique allows more accurate, sensitive, and complete messenger RNA (mRNA) profiling with less potential bias and compromise. RNA-Seq technology has been applied to many plants, including maize, rice, wheat, and soybean [8]. For example, RNA-Seq analysis of the transcriptome of the developing wheat grain has enabled the analysis of genes determining grain functional and nutritional quality (flour yield and bread quality) [19]. Candidate genes that are involved in wheat grain size formation at seed development stages have been identified using RNA-Seq [20]. In maize, transcriptome dynamics during endosperm development has been studied demonstrating the expression of most of the protein coding genes during early endosperm development [21].

A few studies have aimed to better understand the dynamics of key genes and regulatory mechanisms in rice associated with morphology [22], physiology [23], abiotic and biotic stress mechanisms [24,25,26]. Transcriptional control of nutrients during grain filling in rice has been studied by designing a GeneChip microarray covering half of the rice genome to understand the effect of genes participating in nutrient partitioning on starch quality [27]. Using quantitative real-time RT-PCR, the gene expression of 27 starch synthesis-related genes have been evaluated in *O. sativa* seeds at different times after flowering [28]. Gene expression dynamics of peduncle development at heading stages in two *indica* rice cultivars were also investigated using RNA-Seq data [22]. Molecular and cellular events associated with rice embryogenesis at three embryo development stages: 3–5 DAP (days after pollination), 7 DAP, and 14 DAP have been investigated in the *indica* rice transcriptome [8]). In addition, transcriptional networks controlling rice seed development at low temperature [3] and transcriptional regulation and expression of the key genes involved in rice sheath blight resistance [29] have been assessed by analysis of the transcriptome.

We now report the use of RNA -Seq to better characterize rice seed development. Australian wild rice taxa (*Oryza meridionalis* and *Oryza rufipogon* type taxa) are geographically and genetically distinct from Asian wild rice [30] and have not been genetically impacted by interbreeding with domesticated rice. Consequently, these wild rice taxa are ideal for expression analysis of genes associated with domestication traits. We determined gene expression related to domestication traits at different seed development stages in two Australian wild rice and compared it to corresponding gene expression in domesticated rice. (*Oryza sativa* ssp. *japonica* var. Nipponbare). Starch is the main component of rice seed endosperm. Starch and sucrose metabolism is a vital cellular mechanism and plays a vital role in carbohydrate metabolism in the rice seed endosperm. Therefore, we investigated the expression of genes that are involved in starch and sucrose metabolism and seed storage proteins, the second most abundant components of rice seeds, influencing the nutritional quality, pasting, and textural properties of cooked rice. We also investigated the expression of genes involved in the regulation of rice seed development. These data may enhance our understanding of the molecular and cellular processes in rice seed development and the impact of domestication on these processes.

## 2. Results

### 2.1. The Mapping Output of RNA-Seq Reads

Quality trimmed RNA-Seq reads were mapped to a reference genome that was constructed with the CDS (coding sequences) of Os-Nipponbare-Reference-IRGSP-1.0 [31], long non-coding RNA (lncRNA), and unmapped transcripts for this study using CLC-GWB version 20 software (CLC Bio, Aarhus, Denmark). The total number of mapped reads for *O. meridionalis* ranged from 34,216,083 to 68,554,210, for *O. rufipogon* type taxa from 46,419,022 to 72,530,370, and for *O. sativa* from 40,919,710 to 67,218,116 (Appendix A). The percentage of mapped reads for *O. meridionalis* ranged from 42.4% (25DPA-R1) to 83.6% (5DPA-R3). The percentage of mapped reads for *O. rufipogon* type taxa varied from 69.8% (25DPA-R3) to 81.8% (10DPA-R1). The percentage of mapped reads for *O. sativa* ranged between 72.4% (15DPA-R2) and 80.3% (5DPA-R3). In all species, RNA-Seq reads from early DPAs showed the highest mapping percentages, however this rate decreased with the maturation of the seeds.

### 2.2. Analysis of Differentially Expressed Genes (DEGs)

Determination of the differentially expressed genes remains the core objective of many gene expression analyses. A total of ten comparison groups for *O. meridionalis*, six comparison groups for *O. rufipogon* type taxa, and three comparison groups for *O. sativa* were made to determine differentially expressed genes at various seed developing stages. The number of genes that were significantly differentially expressed varied in all comparison groups (Figure 1). DEGs were considered significant at an FDR *p*-value ≤ 0.01. The positive and negative fold change values were the basis for the identification of up-regulation and down-regulation of DEGs, respectively. In *O. meridionalis*, the total number of DEGs was highest between 25 DPA vs. 5 DPA comparison group (5729), followed by 4864 between 20 DPA vs. 5 DPA, 1370 between 15 DPA vs. 5 DPA, and 1048 between 10 DPA vs. 5 DPA where 5 DPA was a control group (Figure 1a). The other comparison groups showed a comparatively low number of significant DEGs such that only 31 transcript sequences were differentially expressed between 25 DPA vs. 20 DPA, inferring the number of DEGs decreased with seed maturation. The highest number of transcript sequences (3459) were upregulated between 25 DPA vs. 5 DPA and the lowest number of upregulated transcript sequences (20) was realized for 25 DPA vs. 20 DPA. The highest number of transcript sequences (3066) were determined as downregulated DEGs between 20 DPA vs. 5 DPA while for the 25 DPA vs. 20 DPA comparison group accounted for the lowest (11), inferring most of the DEGs were upregulated at 5 DPA. In *O. rufipogon* type taxa, the highest number of transcript sequences (41,909) were differentially expressed in 15 DPA vs. 5 DPA comparison group, and therefore, the highest number of upregulated DEGs was 34,686 between 15 DPA vs. 5 DPA (Figure 1b). Noticeably, a very low number of DEGs (92) accounted for 15 DPA vs. 10 DPA with the lowest number of upregulated (35) and downregulated sequences (57). Like *O. meridionalis*, the highest number of transcript sequences were differentially expressed between 25 DPA vs. 5 DPA (12,674), followed by 1690 for 15 DPA vs. 5 DPA, and 1152 for 25 DPA vs. 15 DPA in *O. sativa* (Figure 1c). The number of downregulated DEGs (7123) was higher than upregulated DEGs (5551) for 25 DPA vs. 5 DPA, which was different from the same comparison group for *O. meridionalis* and *O. rufipogon* type taxa.

### 2.3. Expression Patterns of Genes Associated with Domestication Traits

The gene expression of four seed-related domestication traits including seed shattering, grain size, hull colour, and pericarp colour and seed coat were investigated based on RNA-Seq analysis and expressed as the reads per kilobase of transcript per million reads mapped (RPKM) values. The measure RPKM is a with-in sample normalization method that eliminates the feature-length and library size effects and therefore, is widely used in gene expression analyses [9,32]. Gene expressed based on an average RPKM (± standard deviation) of the replicates was determined for domestication genes at five seed development stages of *O. meridionalis*, four seed development stages of *O. rufipogon* type taxa and three seed development stages of *O. sativa*. The gene expression associated with domestication traits varied across all the seed development stages of all the species (Appendix A). The pattern of gene expressed at the three common seed development stages (5 DPA, 15 DPA, and 25 DPA) for all the species is shown in Figure 2. Expression of the *qSH1*(*Shattering (QTL)-1*)) locus was high at 5 DPA and declined with the seed maturation in both wild taxa (Figure 2a). Expression patterns of the *SH4* (*shattering 4*) gene was similar to the *qSH1* gene, where the *SH4* expression declined throughout seed development in the wild rices but was constant in the domesticated rice (Figure 2b). However, *qSH1* and *SH4* loci were not significantly differentially expressed at any of the comparison groups of seed development stages in all the species. The expression of another major seed shattering gene *OsSh1* (*shattering 1*) was high at early stage (5 DPA) of seed development and declined thereafter during seed development for all the species (Figure 2c). Expression of this gene was significantly upregulated at 5 DPA when compared to expression at 25 DPA (Appendix A).

High expression of the *grain size 5* (*GS5*) gene was observed at 5 DPA in both wild rice taxa and *O. sativa* (Figure 2d). The *GS5* gene was not differentially expressed in any of the comparison groups in *O. meridionalis*, however it was differentially expressed at 5 DPA in *O. rufipogon* type taxa (15 DPA vs. 5 DPA and 25 DPA vs. 5 DPA) and *O. sativa* (25 DPA vs. 5 DPA) (Appendix A). The gene expression of *grain size 3* (*GS3*) was high at 15 DPA in *O. rufipogon* type taxa and *O. sativa*, however the gene expression reached a plateau from 15 DPA to 25 DPA in *O. sativa* (Figure 2e). However, this gene was not significantly differentially expressed in all the species. The *black hull 4* (*Bh4*) gene is known to be responsible for black hull colour in rice [33]. In all the rice species, expression of this gene was not detectable at 5 DPA but was thereafter during seed development (Figure 2f). Compared to the expression at the 5 DPA stage, significant upregulation was observed at all stages during seed development in both the wild rice taxa, and at 15 DPA and 25 DPA stages in *O. sativa* (Figure 2f and Appendix A). *Brown pericarp and seed coat* (*Rc*) gene responsible for brown pericarp [34] was highly expressed at 5 DPA in both *O. meridionalis* and *O. rufipogon* type taxa (Figure 2g). This gene was significantly upregulated at 5 DPA and 10 DPA in both *O. meridionalis* and *O. rufipogon* type taxa and at 5 DPA in *O. sativa* (Appendix A).

### 2.4. Expression Patterns of Genes Associated with Starch and Sucrose Metabolism

Eight important genes (*AGPS1*, *AGPS2*, *GBSSI*, *SSIIa*, *BEIIb*, *ISA1*, *PHOL*, and *SuSy1*) that control starch and sucrose metabolism in rice seeds were investigated based on RNA-Seq analysis and expressed as the reads per kilobase of transcript per million reads mapped (RPKM) values to determine the pattern of gene expression in developing seeds (Appendix A). *ADP-glucose pyrophosphorylase small subunit 1* (*AGPS1*) is involved in starch synthesis in the early stages of endosperm development. The expression of the *AGPS1* gene was high at 5 DPA and declined in the later stages of seed maturation in both wild taxa and domesticated rice (Figure 3a). *ADP glucose pyrophosphorylase small subunit 2* (*AGPS2*) regulates starch synthesis in the middle to late stage of endosperm development. The *AGPS2* gene was highly expressed at 5 DPA in all the species (Figure 3b). *AGPS1* gene was significantly upregulated at 5 DPA between 25 DPA vs. 5 DPA and 20 DPA vs. 5 DPA comparison groups in *O. meridionalis* and *O. sativa* (25 DPA vs. 5 DPA) while it was significantly upregulated at both 5DPA (15 DPA vs. 5 DPA, 25 DPA vs. 5 DPA) and at 10 DPA (25 DPA vs. 10 DPA) in *O. rufipogon* type taxa. The *AGPS2* gene was differentially expressed at 5 DPA in a comparison group between 25 DPA vs. 5 DPA with fold change value −4.18 and −2.26 in *O. meridionalis* and *O. sativa*, respectively where 5 DPA was a control group (Appendix A). The *granule-bound starch synthase I* (*GBSSI*) gene responsible for the synthesis of amylose showed a high level of gene expression at 10 DPA to 15 DPA in both wild taxa and remained high throughout the seed development stages. While the gene expression of *GBSSI* was very low in *O. sativa* throughout the seed development (Figure 3c). However, *GBSSI* was not differentially expressed in both wild rice taxa, and only significantly upregulated at 5 DPA (25 DPA vs. 5 DPA) in *O. sativa* (Appendix A). The expression of soluble *starch synthase IIa* (*SSIIa*), *starch branching enzyme IIb* (*BEIIb*), *isoamylase 1* (*ISA1*), and *starch phosphorylase* (*PHOL*) genes were high at an early stage (5 DPA) of seed development and then started to decline at the later stages (25 DPA) in both wild taxa and domesticated rice (Figure 3d–g). Significant upregulation of these genes was detected at 5 DPA (*SSIIa*, *BEIIb*, *ISA1*, and *PHOL*) and 10 DPA (*PHOL*) in *O. meridionalis* (Appendix A). In the *O. rufipogon* type taxa, these genes were significantly upregulated at both 5 DPA and 10 DPA in the three comparison groups: 15 DPA vs. 5 DPA, 25 DPA vs. 5 DPA, and 25 DPA vs. 10 DPA. In *O. sativa*, *SSIIa*, *BEIIb*, and *PHOL* genes were significantly upregulated at 5 DPA in a comparison group 25 DPA and 5 DPA while the *ISA1* gene was differentially expressed at 5 DPA (25 DPA vs. 5 DPA) and 15 DPA (25 DPA vs. 15 DPA) (Appendix A). *Sucrose synthase 1* (*SuSy1*) is a pivotal cytoplasmic enzyme involved in sugar metabolism. A high level of mRNA transcripts for this gene was observed at 5 DPA in all the species declining with the progression of seed maturation (Figure 3h). This gene was significantly differentially expressed at 5 DPA (25 DPA vs. 5 DPA and 20 DPA vs. 5 DPA), 10 DPA (20 DPA vs. 10 DPA), and 15 DPA (20 DPA vs. 15 DPA) in *O. meridionalis*, 5 DPA (15 DPA vs. 5 DPA and 25 DPA vs. 5 DPA) and 10 DPA (25 DPA vs. 10 DPA) in *O. rufipogon* type taxa, and at 5 DPA (25 DPA vs. 5 DPA) in *O. sativa* (Appendix A).

### 2.5. Expression Patterns of Genes Associated with Seed Storage Proteins

In diploid rice, glutelin proteins are encoded by 18 full-length genes belonging to the four sub-families: *Glutelin A* (*GluA*), *Glutelin B* (*GluB*), *Glutelin C* (*GluC*), and *Glutelin D* (*GluD*) classified based on their amino acid sequences [35]. One gene from each glutelin sub-family was investigated based on RNA-Seq analysis and expressed as the reads per kilobase of transcript per million reads mapped (RPKM) values to understand the expression of seed storage protein genes in developing seeds (Appendix A). *Glutelin type A-1* (*GluA-1*), *Glutelin type B-1a* (*GluB-1a*) and *Glutelin type C-1* (*GluC-1*) genes showed high expression at 5 DPA and declined with the progress of seed development in both wild rice taxa and *O. sativa* (Figure 4a–c). The *GluA-1* and *GluB-1a* genes were not significantly differentially expressed in both wild rice taxa. Negative fold chain values for *GluA-1* (−3.51) and *GluB-1a* (−3.27) genes in a 25DPA vs. 5DPA comparison (5 DPA as a control group) inferred a significant up-regulation at 5DPA in *O. sativa*. *GluC-1* gene was significantly differentially expressed in a 25 DPA vs. 5 DPA comparison group in *O. meridionalis* (−3.92), *O. rufipogon* type taxa (−2.55), and *O. sativa* (−2.40). *Glutelin type D-1* (*GluD-1*) gene from *glutelin type D* sub-family was high at 5 DPA in both wild rice taxa except for *O. sativa* where the gene expression reached a peak at 15 DPA, and then gradually decreased during seed development (Figure 4d). However, the *GluD-1* gene was not differentially expressed at any of the comparison groups of all the species.

### 2.6. Discovery of Novel Transcripts

The BLAST (Basic Local Alignment Search Tool) analysis of the newly constructed reference genome including coding sequences (CDS) of the Os-Nipponbare-Reference-IRGSP-1.0 genome, long non-coding RNA (lncRNA) sequences and unmapped transcripts from the three species revealed a total of 38 novel transcripts with coding potential corresponding to domestication genes, starch and sucrose metabolism, and seed storage proteins (Appendix A; Sequence data provided as Appendix A). These novel transcripts had four types of coding potential based on start and stop signals: (1) complete transcript with both start and stop codon, (2) 3′partial transcript with missing stop codon and presumably part of the C-terminus, (3) 5′partial transcript with missing start codon and presumably part of the N-terminus, and (4) internal transcript with both start and stop codons missing. Of 38 novel transcripts, one unmapped transcript from *O. rufipogon* type taxa at 25 DPA had an annotated gene ontology (GO) term for the *black hull 4* (*Bh4*) gene, with a length of 823 bp (5′ partial transcript). A total of 28 unmapped transcripts had annotated GO terms for starch and sucrose metabolism-related genes. Of these, two transcripts were determined for *granule-bound starch synthase I* (*GBSSI*; 160–286 bp), followed by one transcript for *granule-bound starch synthase II* (*GBSSII*; 192 bp), four transcripts for *soluble starch synthase I* (*SSI*; 164–2985 bp), three transcripts for *soluble starch synthase IIIa* (*SSIIIa*; 200–279 bp), three transcripts for *soluble starch synthase VIa* (*SSIVa*; 212–306 bp), three transcripts for *starch branching enzyme I* (*BEI*; 234–361 bp), three transcripts for *starch branching enzyme IIa* (*BEIIa*; 216–550 bp), one transcript for *starch branching enzyme IIb* (*BEIIb*;186 bp), one transcript for *isoamylase 1* (*ISA1*; 477 bp), one transcript for *pullulanase* (*PUL*; 2051 bp), two transcripts for *starch phosphorylase* (*PHOL*; 196–390 bp), two transcripts for *sucrose synthase 2* (*SuSy2*; 209–234 bp), one transcript for *sucrose synthase 3* (*SuSy3*; 184 bp), and one transcript for *sucrose synthase 6* (*SuSy6*; 163 bp). A total of 9 unmapped transcripts had annotated GO terms for seed storage proteins-related genes. The seed storage protein *glutelin* gene was represented by three transcripts (172–320) bp, followed by two transcripts for *glutelin type-A 1* (*GluA-1*; 172–321 bp), one transcript for *glutelin type-A 2* (*GluA-2*; 343 bp), and three transcripts for *glutelin type-B 5* (*GluB-5*; 233–315 bp). Most of the novel transcripts showed internal type of coding potential with start and stop codon missing. Surprisingly, all the novel transcripts identified for domestication genes, starch and sucrose metabolism, and seed storage proteins were derived from the mature stage of seed development (25 DPA) in *O. rufipogon* type taxa and *O. sativa*, inferring transcripts from the mature stages of seed might be lacking in the current reference genome.

### 2.7. Functional Annotation of DEGs

Gene ontology (GO) analysis was applied to functionally categorize the differentially expressed genes (DEGs) into three groups: biological process (BP), cellular component (CC), and molecular function (MF). Among the 8510 significantly differentially expressed genes at five seed developing stages in *O. meridionalis*, 5471 genes had significant hits in the GO protein database and 3155 genes had enzyme code annotation. Of 55,491 DEGs of *O. rufipogon* type taxa, 19,618 genes, and 10,134 genes had significant GO hits and enzyme code annotation, respectively. In *O. sativa*, a total of 14,062 genes were determined from three seed developing stages of which 8814 had GO hits and 4719 had enzyme code annotation. In each functional category of the GO classification, there were 1490 GO terms for BP, 365 GO terms for CC, and 1261 for MF in *O. meridionalis*. Similarly, BP, CC, and MF categories of GO classification in *O. rufipogon* type taxa represented 2263, 653, and 1787 GO terms, respectively. In *O. sativa*, BP, CC, and MF accounted for 1953, 487, and 1430 GO terms, respectively. The top ten functional groups from each GO classification for each species were presented in Appendix A. In all species, organic substance metabolic process (GO:0071704) with 2895 for *O. meridionalis*, 9773 genes for *O. rufipogon* type taxa, and 4785 genes for *O. sativa* were dominant in the biological process category. Like BP, organic cyclic compound binding (GO:0097159) and intracellular anatomical structure (GO:0005622) were the most dominant in MF and CC categories of GO classification, representing 2116 genes and 2010 genes, respectively for *O. meridionalis*, 7595 and 8010 genes, respectively for *O. rufipogon* type taxa, and 3577 and 4509 genes, respectively for *O. sativa*.

KEGG pathway analysis was conducted to identify the active biological processes in the developing seeds (Figure 5). The highest number of DEGs (8739) were found to be linked with 334 pathways from five KEGG pathways categories (genetic information processing, cellular process, environmental information processing, organismal systems, and metabolism) in *O. rufipogon* type taxa compared to *O. meridionalis* and *O. sativa*. A total of 2804 DEGs were linked to 319 pathways in *O. meridionalis* while 3729 DEGs were found in 326 pathways related to the three seed developing stages of *O. sativa*. The most common pathways found in all taxa included ribosome (ko0310), starch and sucrose metabolism (ko00500), glycolysis/gluconeogenesis (ko00010), oxidative phosphorylation (ko00190), and biosynthesis of cofactors (ko01240). The ribosome (ko0310) pathway was the most dominant pathway in wild taxa while the starch and sucrose metabolism pathway (ko00500) was dominant in *O. sativa*. The highest number of DEGs from *O. rufipogon* type taxa and *O. meridionalis* was linked to the ribosome pathway (ko0310), accounting for 742 and 222, respectively. Similarly, the highest number of DEGs (218) was linked to starch and sucrose metabolism pathway (ko00500) in *O. sativa*. 

## 3. Discussion

Transcriptome analysis revealed the highest number of genes are differentially expressed between 25 DPA and 5 DPA comparison groups in *O. meridionalis*, between 15 DPA and 5 DPA in *O. rufipogon*, and between 25 DPA and 5 DPA in *O. sativa* when 5 DPA is a control group. Consequently, the maximum number of DEGs are upregulated between 25 DPA vs. 5 DPA and downregulated between 20 DPA vs. 5 DPA in *O. meridionalis*, inferring most of the DEGs are expressed from early (5 DPA) to late stages (25 DPA) of seed development. While in *O. rufipogon* type taxa, up-regulation, and down-regulation of the maximum number of DEGs occur between 15 DPA vs. 5 DPA and 25 DPA vs. 5 DPA, respectively, inferring most of the DEGs are expressed at early (5 DPA) to middle (15 DPA) stages of seed development. However, the ratio of upregulated and downregulated DEGs in *O. sativa* was 44% and 56%, respectively in the comparison group between 25 DPA vs. 5 DPA, indicating most of the genes were expressed at an early to late stages of seed development. An earlier transcriptome study of the *indica* rice embryo suggested most of the DEGs were captured between 3–5 and 7 DAP (days after pollination) compared to a comparison group between 7 DAP and 14 DAP [8]. Gene expression declines substantially as the seed matures.

Domestication genes related to seed shattering, grain size, and pericarp colour and seed coat show a common pattern of gene expression in all the species. The gene expression for *qSH1* and *SH4* genes is high at early stages, however these loci are not significantly differentially expressed for the three species. *qSH1* gene encoding a *BEL1-type homeobox* gene involves the formation of an abscission zone at the base of the rice grain [36]. A single base substitution (G to T) at 11,841 bases upstream in the 5′ regulatory region causes a non-seed shattering nature due to the absence of abscission layer formation in *O. sativa* seeds [36]. *SH4* gene encoding Myb3 DNA binding domain, and a nuclear localization signal involves the establishment of the abscission layer at the initial stage but in the mature stage it activates the abscission process at the junction of flower and pedicel wherein mature seeds tend to get separated from their mother plants [37]. Unlike *qSH1* and *SH4*, the *OsSh1* locus encoding YABBY transcription factor responsible for seed shattering is significantly differentially expressed at the early stage of seed development (5 DPA) in both wild rice taxa and *O. sativa*. A recent study reveals the expression of *qSH1* does not affect the loss of the function of the *OsSh1* locus in the young panicles of domesticated rice as this gene might act downstream of a *qSH1* gene [38]. The patterns of gene expression of *qSH1*, *SH4*, and *OsSh1* suggest that wild rice may start accumulating transcripts in the abscisic layer at an early stage of seed development leading to shattering in wild rice taxa while the dysfunction of these genes may produce an incomplete abscisic zone at the middle to later stages of seed development in *O. sativa*.

The abundance of mRNA transcripts of *GS5* locus related to grain size is high at 5 DPA in all the species, however this gene was significantly upregulated at 5 DPA in *O. rufipogon* type taxa and *O. sativa*. This gene encodes a putative serine carboxypeptidase that acts as a positive regulator of grain size. Therefore, the higher expression of this gene increases cell division resulting in increases in grain width and yield [39]. The grain size of *O. rufipogon* type taxa are medium (5.81–5.90 mm) and *O. meridionalis* exhibits long-grain (6.20–6.80 mm) [40]. Likewise, the *O. rufipogon* type taxa, medium-size grain is a feature of *O. sativa* [41]. In contrast, the expression of the *GS3* gene increases from 5 DPA to 25 DPA in *O. sativa*. However, the level of transcripts of this gene was low in both wild rice taxa. Notably, this gene was not differentially expressed in both wild rice taxa and *O. sativa*. However, an earlier study showed that *GS3* is highly expressed in young panicles and decreased with panicle development in the long grain *indica* rice variety Minghui 63 [41]. *GS3* encodes the γ-subunit 3 of a heterotrimeric G-protein and acts as a negative regulator of grain size, inferring the wild type allele corresponds to medium grain and a mutation result in large grain size [42]. However, an in-frame insertion of 3 bp (*GS3-2*) in the fifth exon did not change the grain length in *O. sativa* which is a medium-size grain cultivar [41]. In wheat, most of the grain size-related genes are expressed at 14 DAP and are likely to be linked with starch and sucrose metabolism [20].

Black hull colour (*Bh4*) is one of the domestication loci that have been changed in the domestication process. This locus is located on rice chromosome 4 encoding an amino acid transporter. The locus is significantly upregulated from an early stage (10 DPA) to later stage (25 DPA) of seed development in both wild rice taxa and from middle stage (15 DPA) to later stage (25 DPA) of seed development in *O. sativa*. This is consistent with Zhu et al. [43] as they suggest that this locus is only expressed in the maturing hull. The hull colour of mature grains of *O. meridionalis* and *O. rufipogon* type taxa are black [40]. A transition from black hull colour to straw-white hull colour in domesticated rice (*O. sativa* ssp. *indica* cv. Guangluai 4) results from a 22-bp deletion within exon 3 of the *Bh4* gene interrupting the function of *Bh4* [43].

While *brown pericarp and seed coat* (*Rc*) locus was significantly upregulated at 5 and 10 DPA in both wild species but only at 5 DPA in *O. sativa*. The grain colour of *O. meridionalis* and *O. rufipogon* type taxa is red-brown and dark brown, respectively [40]. *Rc* gene encodes a basic helix-loop-helix (bHLH) protein regulating red pericarp in wild rice. A frameshift deletion of 14 bp at exon 6 in this locus resulted in an early termination codon that truncated the protein of the bHLH domain and ultimately produced white grain rice cultivars [34]. This gene is expressed in both red-grained and white-grained rice; however, a shortened transcript is present in white rice varieties [34]. Our study also reveals that the expression of *Rc* gene was very low in white-grained *O. sativa* compared to the wild rice taxa.

Starch is stored in the highest concentration in the starchy endosperm. The rice endosperm starch biosynthetic pathway is one of the most meticulously described metabolic pathways in plants. Starch is an important domestication trait that has been under strong selection during rice domestication [44]. Evidence of positive selection has been postulated for *AGPL2* in *O. rufipogon*, *GBSSI* in temperate *japonica* and tropical *japonica* varieties, and *GBSSI* and *BEIIb* in aromatic varieties [44]. The gene expression pattern of seven key starch synthesis genes (*AGPS1*, *AGPS2*, *GBSSI*, *SSIIa*, *BEIIb*, *ISA1*, and *PHOL*) were investigated in this study. These key genes can be divided into five classes of enzyme: ADP-glucose pyrophosphorylase (*AGPS1*, *AGPS2*), starch synthase (*GBSSI*, *SSIIa*), starch branching enzyme (*BEIIb*), starch de-branching enzyme (*ISA1*), and starch phosphorylase (*PHOL*). This study reveals that all the starch synthesis-related genes are highly expressed at the early to middle stage of seed development and decrease with the progress of seed maturation. *AGPS1* and *AGPS2* play an important role in starch accumulation by AGP-glucose synthesis in rice endosperm in the early stage and middle to later stages of endosperm development, respectively. The study shows the abundance of mRNA transcripts at the early stage for *AGPS1* and early to middle stages of seed development for *AGPS2*, however they were significantly upregulated only at an early stage in all the species (except for *AGPS2* in *O. rufipogon* type taxa). *GBSSI* gene which plays a key role in the production of amylose in the pathway, is only significantly upregulated in the early stage (5 DPA) of seed development in *O. sativa*. Gene from soluble starch synthase *(SSIIa*), starch branching (*BEIIb*), starch debranching (*ISA1*), and starch phosphorylase (*PHOL*) enzyme group show an increased pattern of expression at an early stage (5 DPA) to middle stage (15 DPA) of seed development. This result seems relevant as starch accumulation in the endosperm of rice seeds increases from 4 to 18 days after anthesis [45]. Ohdan et al. [28] studied the expression profiling of starch synthesis-related genes in *O. sativa* using quantitative real-time RT-PCR and suggested that *AGPS1* reaches a peak at 3 to/or 5 days after flowering (DAF) while *AGPS2*, *GBSSI*, *SSIIa*, *BEIIb*, and *ISA1* rapidly increase in around 5–7 DAF and *PHOL* reaches at peak at 5 DAF. Similar pattern of gene expression has been also observed in other cereals crops like wheat. In wheat, most of the transcripts associated with starch and seed storage proteins have been expressed at 14 DPA [19]. The sucrose synthase enzyme (sucrose-synthase 1, EC:2.4.1.13) is an important source of a sucrose-cleaving enzyme that provides UDP-glucose and fructose during sucrose metabolism. This study reveals that *SuSy1* expression was significantly upregulated from early to middle stages of seed development in both wild rice taxa while the high expression of this gene was limited to the early stage in *O. sativa*. A previous study also suggests that *SuSy1* is expressed largely in the milky stage of rice endosperm with the highest expression at 3–11 days after pollination [46].

Seed storage proteins are the second most abundant components of rice seeds after starch, accounting for approximately 7–10% of the seed weight, and are key factors influencing the nutritional quality, pasting, and textural properties of cooked rice [35]. The mRNA transcripts for seed storage proteins (*GluA-1*, *GluB-1a*, *GluC-1*, and *GluD-1*) start to increase from 5 DPA and drop at the later stage of seed development, except for *GluD-1* in *O. sativa*. The transcripts accumulation for *GluD-1* was very low in *O. meridionalis* throughout the seed development stages. *GluA-1*, *GluB-1a*, *GluC-1* genes were significantly upregulated at early stages (5 DPA) in *O. sativa* while *GluC-1* was differentially expressed at early stages in both wild rice taxa. Kawakatsu et al. [47] have evaluated the gene expression of *GluD-1* using northern blotting in seed tissue of *O. sativa* and revealed that this gene was accumulated at 5 DPA and gradually increased until 30 DAF. In contrast, the previous study reported that the expression of rice glutelin genes in diploid rice started at 4–6 DAF, reached a maximum level at 10–14 DAF, and afterward decreased [48]. Recent qRT-PCR analysis of *glutelin* gene (*GluA-1*, *GluA-2*, *GluB-1*, *GluB-2*, *GluC-1*, and *GluD-1*) in developing seeds of an *indica* rice cultivar (9311-2x) reveal the expression of these genes started to increase at 5 DAF, reached a peak at 17 DAF and gradually declined from 21 DAF [35].

A BLAST search of the new reference genome sequences generated for this study reveals a substantial number of novel transcripts (38 transcripts) with coding potential related to domestication genes, starch and sucrose metabolism, and seed storage proteins. These transcripts were identified in the mature seed tissue (25 DPA) in *O. rufipogon* type taxa and *O. sativa*, which indicates that they might not have been incorporated in the current rice reference genome. Information on these novel transcripts may provide new insight into the starch synthesis pathway.

Gene ontology analysis reveals organic substance metabolic process (GO:0071704) was predominant in all species while organic cyclic compound binding (GO:0097159) and intracellular anatomical structure (GO:0005622) were the most dominant in molecular function and cellular component, respectively. The highest numbers of DEGs were for ribosomes and starch and sucrose metabolism pathways. An earlier study postulates that the largest numbers of genes were involved in the cellular and metabolic regulatory pathway and were expressed in the early and middle stages of embryogenesis [8]. The highest number of DEGs were linked to ribosome and starch/sucrose metabolism in all the species, which is consistent with the findings by Xu et al. [8].

This study reveals a distinctive pattern of gene expression in the developing seeds of wild and domesticated rice. A substantial number of novel transcripts associated with domestication traits, starch and sucrose metabolism and seed storage proteins were identified. These additional novel transcripts may be of interest to improve the current reference genome as well as providing new insights into starch and sucrose metabolism in rice. The use of wild rice as a source of novel variation will also be aided by this information on the expression of genes in wild and domesticated rice that have the potential to control seed composition and quality. 

## 4. Materials and Methods

### 4.1. Plant Materials

Two Australian wild rice taxa namely, *O. meridionalis* and *O. rufipogon* type taxa, and one *japonica* rice cultivar (*O. sativa* ssp. *japonica* var Nipponbare) were investigated. The two wild rice taxa are distinct and novel AA genome taxa in Northern Queensland, Australia [49]. The whole panicle was collected from seed development stages at days post anthesis (DPA) from *O. meridionalis* (5 DPA, 10 DPA, 15 DPA, 20 DPA, and 25 DPA), *O. rufipogon* type taxa (5 DPA, 10 DPA, 15 DPA, and 25 DPA), and *O. sativa* (5 DPA, 15 DPA, and 25 DPA). Panicles from 20 DPA were not collected for *O. rufipogon* type taxa due to the limited availability of panicles of this relatively rare type. It was noticed that rice panicles took approximately 5–7 days to complete anthesis since the emergence of heads. Therefore, each DPA was estimated after 5 days of the emergence of the panicle. Each panicle was enclosed with a cellophane perforated bag to avoid cross-fertilization and tagged with the date of panicle emergence to estimate the DPA correctly. The whole experiment was performed in triplicates, except for *O. rufipogon* type taxa (quadruplets in 5 DPA, 10 DPA, and 15 DPA). These plants were grown in the glasshouse at The University of Queensland, Australia and panicles were collected from each replicate, separately. The whole panicles were snap-frozen with liquid nitrogen immediately after collection and stored at −80 °C at the laboratory. Empty seeds without endosperm were discarded from the collections before pulverizing the seeds. Seeds were pulverized for each DPA per replicate from each species, separately using Qiagen Tissue Lyser II (Qiagen, Germantown, Maryland, United States of America) under cryogenic condition and stored at −80 °C until RNA extraction.

### 4.2. RNA Extraction and Purity Determination

The seed endosperm contains a high level of starch, which interferes with the RNA extraction process as starch tends to precipitate [50]. Therefore, to screen out the suitable RNA extraction protocol for the extraction of RNA from wild and domesticated rice seeds, an RNA extraction protocol was optimized with a combination of CTAB and QIAGEN RNeasy plant mini kit (QIAGEN, Inc., Valencia, CA, USA) that efficiently produced good quality and quantity of RNA from seed tissues. The CTAB method was slightly modified from Wang and Stegemann (2010) and incorporated with and QIAGEN RNeasy plant mini kit for pure RNA yield. The quality, quantity, and integrity of RNA were checked with a NanoDrop8000 spectrophotometer (Thermo Fisher Scientific, Wilmington, DE, USA) and an Agilent Bioanalyzer 2100 with the Agilent RNA 6000 Nano kit (Agilent Technologies, Santa Clara, CA, USA). The 260/280 nm ratio from the NanoDrop8000 spectrophotometer ranged from 1.57 to 2.13. The ratio of absorbance at 260 nm and 280 nm is used to assess the purity of RNA. A ratio of ~2.0 is generally regarded as pure RNA. RNA integrity numbers (RIN) obtained from Agilent Bioanalyzer 2100 ranged from 5.1 to 9.2. A RIN score of 8 is desirable and below 7.5 is contamination compromised in an RNA-Seq experiment.

### 4.3. cDNA Library Preparation, RNA Sequencing, and Quality Check

The complementary DNA (cDNA) library preparation was done with the TruSeq DNA PCR-Free kit following the TruSeq DNA PCR-free sample preparation guide, Part # 15036187 Rev. D (Macrogen Inc., Seoul, Republic of Korea. These libraries were sequenced on a Hisequation 2000 with a read length of 100 bp (Illumina) (Macrogen Inc., Seoul, Republic of Korea). RNA-Seq reads were trimmed at 0.05 and 0.01 quality scores to truncate low-quality read using QIAGEN CLC Genomics Workbench (CLC-GWB) version 20 software (CLC Bio, Aarhus, Denmark). Based on the percentage of loss of bases and reads, 0.01 trimmed reads were designated for further analysis. The total number of raw reads for *O. meridionalis*, *O. rufipogon* type taxa, and *O. sativa* ranged between 59,005,792 and 89,860,934, 60,336,822 and 101,983,884, and 57,326,794 and 86,174,334, respectively. After quality filtering, the total number of 0.01 trimmed reads ranged from 57,646,086 to 87,398,528 in *O. meridionalis*, from 58,851,833 to 99,355,437 in *O. rufipogon* type taxa, from 54,964,733 to 83,679,467 in *O. sativa* (Appendix A).

### 4.4. Construction of a New Reference Genome and RNA-Seq Analysis

Generation of a new reference genome comprises the following steps: (1) mapping of RNA-Seq reads to the coding sequences of Os-Nipponbare-Reference-IRGSP-1.0 genome [31], (2) mapping the unmapped reads to ribosomal RNA (rRNA) database [51], plant long non-coding RNA (lncRNA) database [52], and the 3′ and 5′ UTRs (untranslated regions) of Os-Nipponbare-Reference-IRGSP-1.0 genome; (3) assemblage of unmapped reads into contigs using DE NOVO Assembly; (4) reduction of the redundancy of the unmapped transcripts; (5) functional annotation of unmapped transcripts, (6) prediction of the coding potential of unmapped transcripts, and (7) generation of a new reference genome.

In the first step, RNA-Seq reads from 39 samples were mapped to the coding sequences (CDS) of the reference genome Os-Nipponbare-Reference-IRGSP-1.0, which was downloaded from the Rice Annotation Project Database [31] using RNA-Seq analysis tool in CLC-GWB version 20 software following the parameters: reference type = one reference sequences; use spine-in control = no; mismatch cost = 2; insertion cost = 3; deletion cost = 3; length fraction = 0.8; similarity fraction = 0.8; global alignment = no; strand specific = both; library type = bulk; maximum number of hits for a read = 20, count paired reads as two = yes; expression value total counts. The percentage of mapped reads to CDS ranged between 10.04% (20DPA-R2) and 58.27% (5DPA-R3) for *O. meridionalis*, 13.71% (25DPA-R2) and 57.86% (5DPA-R4) for *O. rufipogon* type taxa, and 31.07% (15DPA-R2) and 55.72% (5DPA-R1) for *O. sativa*. On contrary, the percentage of unmapped reads to CDS ranged between 41.73% (5DPA-R3) and 89.96% (20DPA-R2) for *O. meridionalis*, 42.14% (5DPA-R4) and 86.29% (25DPA-R2) for *O. rufipogon* type taxa, and 44.28% (5DPA-R1) and 68.93% (15DPA-R2) for *O. sativa* (Appendix A). The percentages of mapped reads were low compared to the percentages of unmapped reads for all the species.

In the second step, the high percentage of unmapped reads led us to characterize the unmapped reads. To do so, reads from each developmental stage and replicate were pooled for each species and subsequently mapped to CDS of the *O. sativa* genome, ribosomal RNA (rRNA) database, plant long non-coding RNA (lncRNA) database, and the 3′ and 5′ UTRs (untranslated regions) of Os-Nipponbare-Reference-IRGSP-1.0 genome separately using the tool “Map Reads to Reference” in CLC-GWB version 20 software, with the following parameters: masking mode = no masking; match score = 1; mismatch cost = 2; cost of insertions and deletions = linear gap cost; insertion cost = 3; deletion cost = 3; length fraction = 0.8; similarity fraction = 0.8; global alignment = no; auto-detected paired distances = yes; non-specific match handling = map randomly. In *O. meridionalis*, 28.94% of the total reads mapped to CDS and 71.06% of reads remained unmapped. Of these unmapped reads, 3.46% of reads mapped to rRNA followed by 28.09% to lncRNA and 9.95% to 3′ and 5′ UTRs (untranslated regions) of the *O. sativa* reference genome. However, a substantial percentage of reads (29.56%) of the total was unmapped which might be novel CDS, novel 3′UTRs and 5′UTRs. Similarly, a large proportion of the total reads did not map to CDS, rRNA, lncRNA, and 3′ and 5′ UTRs of the *O. sativa* reference genome, amounting to 20.22% in *O. rufipogon* type taxa and 16.94% in *O. sativa* (Appendix A). As the percentage of mapped reads to lncRNA was quite high in the three species, therefore a total of 19 lncRNA sequences were selected based on coverage ≥ 70 (Appendix A). Coverage was calculated as (consensus length/reference length) × 100.

In the third step, these unmapped reads after mapping to CDS, rRNA, lncRNA, and 3′ and 5′ UTRs of the *O. sativa* reference genome, unmapped reads were assembled into contigs using DE NOVO Assembly tool in CLC-GWB version 20 software with the parameters: automatic bubble size = yes; minimum contig length = 150 bp; automatic word size = yes; performing scaffolding = yes; auto-detect paired distances = yes. In *O. meridionalis*, the total bases of the scaffold were 114,971,604 bp. The total number of contigs was 324,347 with N50 (50% of the total contig length) of 385 bp and the contig length ranged from 150 bp to 35,608 bp. In *O. rufipogon* type taxa, the total bases of the scaffold were 124,259,805 bp. The total number of contigs was 347,810 with N50 of 397 bp and the contig length ranged between 150 bp and 34,535 bp. For *O. sativa*, the total bases of the scaffold were 78,738,384 bp and the total contigs were 223,596 with N50 of 382 bp. The contig length varied between 150 bp and 17,096.

In the fourth step, to reduce sequence redundancy and improve the performance of the sequences, the clustering tool in OmicsBox 1.3.11 (https://www.biobam.com/) was applied in a total of 895,753 unmapped transcripts derived from the DE NOVO assembly analysis of the three species with default parameters, except for sequence identity threshold 0.99. After removing redundancy, a total of 483,274 unmapped transcripts were considered.

In the fifth step, the annotation with gene ontology (GO) terms of these unmapped transcripts was investigated using the Blast2GO functional annotation workflow [53] in OmicsBox version 1.3.11 software (https://www.biobam.com/). Unmapped transcripts were first blasted against the non-redundant protein sequences (nr v5) database using the blastx-fast program with a blast expectation value (e-value) of 1.0E-10. While performing the blastx-fast search, sequences were blasted against the Viridiplantae database. The Basic Local Alignment Search Tool (BLAST) result was subjected to gene ontology (GO) mapping [53] and annotation. The Cloud InterProScan was applied to search for protein domains from the EMBL-EBI (EMBL’s European Bioinformatics Institute) databases. Of 483,274 unmapped transcripts, 483,258 transcripts mapped to the InterProScan protein database, 104,894 transcripts had BLAST hits, 62,913 transcripts had mapping and BLAST hits, and 62,273 transcripts had BLAST, mapping, and annotation with GO terms (Appendix A). However, 378,380 transcripts had no BLAST hit.

In the sixth step, the predict coding regions tool based on TransDecoder 5.5.0 software [54] in OmicsBox 1.3.11 was applied with the following parameters: a minimum protein length = 51 bp and strand specific = false to predict open reading frames (ORFs). The captured ORFs are scanned for homology to known proteins through the Pfam database [55] to identify common protein domains. The predicted ORFs encoded for proteins of four types based on start and stop signals: complete with both start and stop codon, 3′partial with missing stop codon and presumably part of the C-terminus, 5′partial with missing start codon and presumably part of the N-terminus, and internal with both start and stop codons missing. Coding potential analysis revealed 84,082 of the total unmapped transcripts with BLAST hits (104,894) had coding potential of which 14,888 unmapped transcripts were complete transcript, followed by 9640 as 3′partial transcript, 18,613 as 5′partial transcript, and 40,941 as internal transcript. Of 378,380 unmapped transcripts without BLAST hits, a total of 96,241 unmapped transcripts showed coding potential in the coding regions representing 16,136 transcripts as complete transcript, 9048 as 3′pratial transcript, 30,895 as 5′partial transcript and 40,162 as internal transcript. As these unmapped transcripts had potential coding regions, these transcripts were incorporated with the CDS of the Os-Nipponbare-Reference-IRGSP-1.0 genome.

In the final step, a new reference genome was constructed with CDS of the Os-Nipponbare-Reference-IRGSP-1.0 genome (42,319 sequences), unmapped transcripts (483,274 sequences), and 19 lncRNA sequences. A new reference genome consists of 525,612 sequences. RNA-Seq reads from 39 samples were finally mapped to a new reference genome using the RNA-Seq analysis tool in CLC-GWB following the above-mentioned parameters.

### 4.5. Gene Expression Analysis

The level of gene expression of each transcript at different seed developing stages was assessed based on RNA-Seq analysis and expressed as the reads per kilobase of transcript per million reads mapped (RPKM) values. RPKM values are useful descriptive measures for the estimation of the expression level of a gene. Transcripts associated with domestication traits, starch and sucrose metabolism and seed storage protein were analysed against the coding sequences of the reference genome Os-Nipponbare-Reference-IRGSP-1.0 [31]. An average RPKM value of the replicates with (±) standard deviation values were plotted against the three common seed development stages (5 DPA, 15 DPA and 25 DPA) for all the species to determine the pattern of gene expression. Domestication genes such as *Shattering (QTL)-1* (*qSH1*; Os01t0848400-01), *shattering 4* (*SH4*; Os04t0670900-01), *shattering 1* (*OsSh1*; Os03t0650000-01), *grain size 5* (*GS5*; Os05t0158500-01), *grain size 3* (*GS3*; Os03t0407400-01), *black hull 4* (*Bh4*; Os04t0460200-01), and *brown pericarp and seed coat* (*Rc*; Os07t0211500-01) were targeted for the gene expression analysis. The level of gene expression was also realized for eight starch and sucrose metabolism related genes such as *ADP-glucose pyrophosphorylase small subunit 1* (*AGPS1*; Os09t0298200-01), *ADP-glucose pyrophosphorylase small subunit 2* (*AGPS2*; Os08t0345800-01), *granule-bound starch synthase I* (*GBSSI*; Os06t0133000-01), *soluble starch synthase IIa* (*SSIIa*; Os06t0229800-01), *starch branching enzyme IIb* (*BEIIb*; Os02t0528200-01), *isoamylase 1* (*ISA1*; Os08t0520900-01), *starch phosphorylase* (*PHOL*; Os03t0758100-01), and *sucrose synthase 1* (*SuSy1*; Os03t0401300-01). Four genes of seed storage proteins belonging to the four seed storage protein family, including *glutelin type A-1* (*GluA-1*; Os01t0762500-00), *glutelin type B-1a* (*GluB-1a*; Os02t0249800-01), *glutelin type C-1* (*GluC-1*; Os02t0453600-01), and *glutelin type D-1* (*GluD-1*; Os02t0249000-01) were investigated to determine the gene expression pattern at different seed developing stages.

### 4.6. Differentially Expressed Genes (DEGs) Analysis

Differential expression analysis requires the gene expression values that should be compared among comparison groups. To proceed with differential gene expression analysis, a total of three metadata was prepared for the differentially expressed genes (DEGs) analysis: (a) 10 comparison groups of five DPAs for *O. meridionalis*, (b) six comparison groups of four DPAs for *O. rufipogon* type taxa, and (c) three comparison groups of three DPAs for *O. sativa* ssp. *japonica* var. Nipponbare (Appendix A). Replicates for each DPA were combined for DEGs analysis in each species. Differentially expressed genes analysis was performed between two different comparison groups with CLC-GWB version 20 software using the differential expression for RNA-Seq tool following the parameters: technology = whole transcriptome RNA-Seq; filter on average expression for FDR correct = yes; test differential expression due to = time point; comparisons = against a control group. Significant differentially expressed genes were determined based on false discovery rate (FDR) *p*-values of ≤ 0.01. Up-regulation and down-regulation of DEGs were determined based on fold change values. DEGs with positive fold change values correspond to up-regulation and the negative fold change values represent down-regulation. The targeted DEGs corresponding to domestication genes, starch and sucrose metabolism, and seed storage protein were determined to investigate the significant differential expression between seed developing stages.

### 4.7. Discovery of Novel Transcripts

New reference genome generated in this study was blasted against the non-redundant protein sequences (nr v5) database using the blastx-fast program with a blast expectation value (e-value) of 1.0 × 10^−10^ in OmicsBox 1.3.11 (https://www.biobam.com/), following the Blast2GO functional annotation workflow [53]. During the blastx-fast search, sequences were blasted against the Viridiplantae database. The BLAST (Basic Local Alignment Search Tool) result was subjected to gene ontology (GO) mapping [53] and annotation. The Cloud InterProScan was applied to search for protein domains from the EMBL-EBI (EMBL’s European Bioinformatics Institute) databases. Novel transcripts associated with domestication genes, starch and sucrose metabolism, and seed storage proteins were searched from the annotated unmapped transcripts. Then, the coding potential of novel transcripts was determined with the predict coding regions tool in OmicsBox 1.3.11 with the following parameters: a minimum protein length = 51 bp and strand specific = false. The predicted ORFs encoded for proteins are of four types based on start and stop signals: complete with both start and stop codon, 3′partial with missing stop codon and presumably part of the C-terminus, 5′partial with missing start codon and presumably part of the N-terminus, and internal with both start and stop codons missing.

### 4.8. Functional Annotation of the Rice Transcriptome

Plant generic gene ontology (GO) annotation of DEGs from three species was determined using the Blast2GO functional annotation workflow in OmicsBox 1.3.11 software [53]. Gene ontology is a gene classification system that has three ontologies: molecular function (MF), cellular component (CC), and biological process (BP). To investigate the biological pathways in which the identified DEGs were involved in seed developing stages in each species, the Kyoto Encyclopedia of Genes and Genomics (KEGG) analysis was applied in OmicsBox 1.3.11 software using KEGG pathway tools with default parameters.

## Figures and Tables

**Figure 1 ijms-23-13351-f001:**
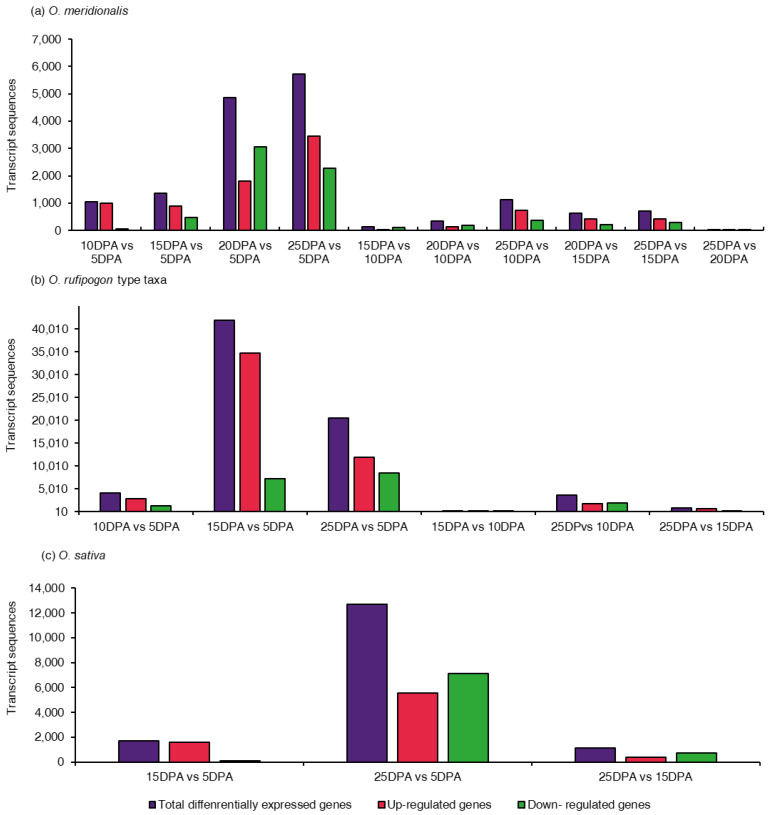
Distribution of differentially expressed genes (DEGs) in comparison groups of the seed developing stages: (**a**) 10 comparison groups from five seed developing stages in *O. meridionalis*; (**b**) six comparison groups from four seed developing stages of *O. rufipogon* type taxa; (**c**) three comparison group from three seed developing stages of *O. sativa*. The *y*-axis represents the number of transcript sequences that were significantly differentially expressed at a false discovery rate (FDR) *p*-value of ≤ 0.01. The *x*-axis represents comparison groups between seed developing stages at days post anthesis (DPA). DPA on the right side of a comparison group represents a control group.

**Figure 2 ijms-23-13351-f002:**
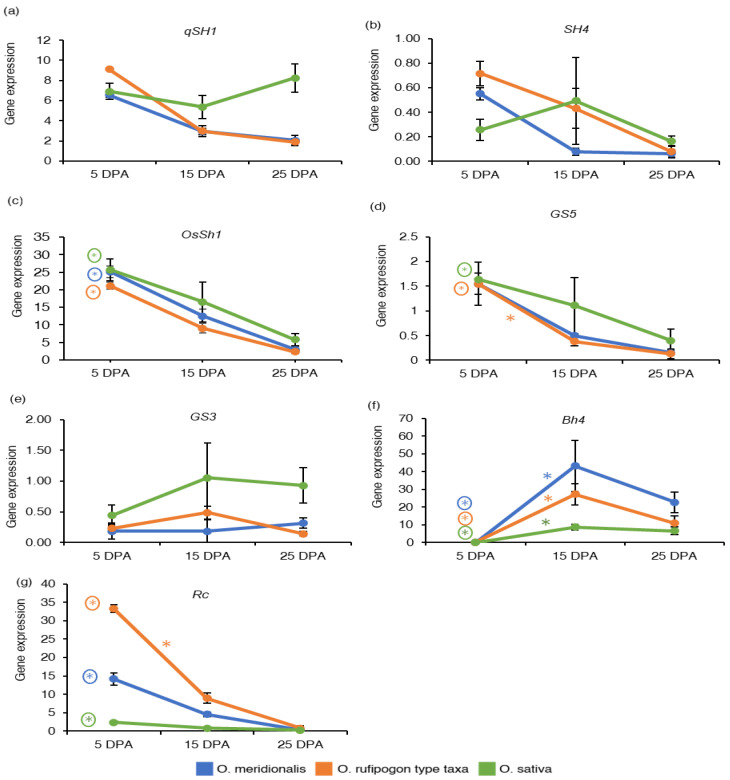
Gene expression of domestication genes: (**a**) *Shattering (QTL)-1* (*qSH1*); (**b**) *shattering 4* (*SH4*); (**c**) *shattering 1* (*OsSh1*); (**d**) *grain size 5* (*GS5*); (**e**) *grain size 3* (*GS3*); (**f**) *black hull 4* (*Bh4*); (**g**) *brown pericarp and seed coat* (*Rc*). The *y*-axis represents gene expression based on RNA-Seq analysis and expressed as reads per kilobase of transcript per million reads mapped (RPKM) values. The *x*-axis represents days post anthesis (DPA). Data are presented as mean ± standard deviation of three biological replicates for *O. meridionalis* and *O. sativa* and four biological replicates for *O. rufipogon* type taxa (except for 25 DPA with three replicates). * symbol above the line between two DPA represents significant differential gene expression and (*) with a circle represents significant differential gene expression between 5 DPA and 25 DPA.

**Figure 3 ijms-23-13351-f003:**
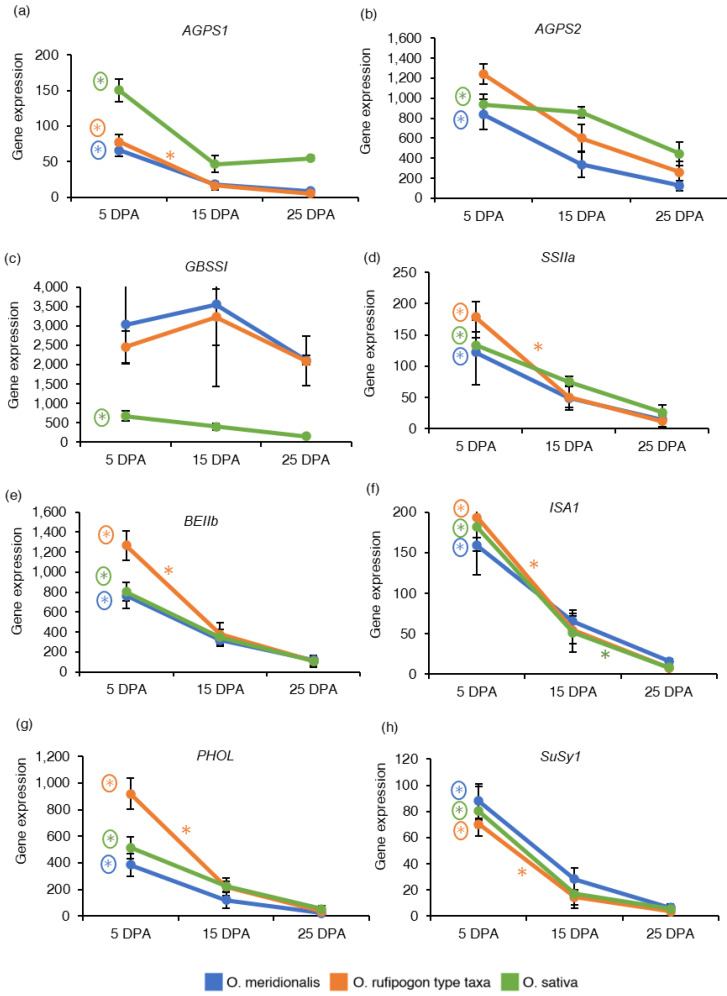
Gene expression of starch and sucrose metabolism-related genes: (**a**) *ADP-glucose pyrophosphorylase small subunit 1* (*AGPS1*); (**b**) *ADP glucose pyrophosphorylase small subunit 2* (*AGPS2*); (**c**) *Granule-bound starch synthase I* (*GBSSI*); (**d**) *Soluble starch synthase IIa* (*SSIIa*); (**e**) *Starch branching enzyme IIb* (*BEIIb*); (**f**) *Isoamylase 1* (*ISA1*); (**g**) *Starch phosphorylase* (*PHOL*); (**h**) *Sucrose synthase 1* (*SuSy1*). The *y*-axis represents gene expression based on RNA-Seq analysis and expressed as reads per kilobase of transcript per million reads mapped (RPKM) values. The *x*-axis represents days post anthesis (DPA). Data are presented as mean ± standard deviation of three biological replicates for *O. meridionalis* and *O. sativa* and four biological replicates for *O. rufipogon* type taxa (except for 25 DPA with three replicates). * symbol above the line between two DPA represents significant differential gene expression and (*) with a circle represents significant differential gene expression between 5 DPA and 25 DP.

**Figure 4 ijms-23-13351-f004:**
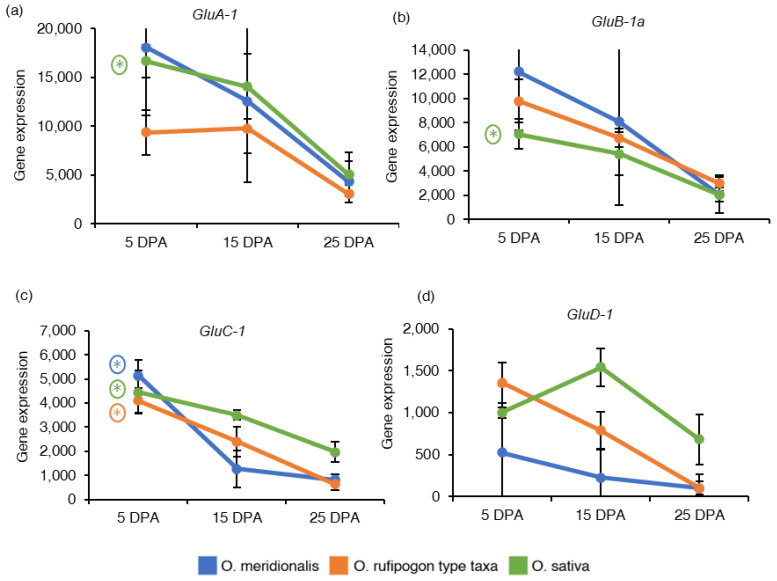
Gene expression of seed storage proteins related genes: (**a**) *Glutelin type A-1* (*GluA-1*), (**b**) *Glutelin type B-1a* (*GluB-1a*), (**c**) *Glutelin type C-1* (*GluC-1*), and (**d**) *Glutelin type D-1* (*GluD-1*). The *y*-axis represents gene expression based on RNA-Seq analysis and expressed as reads per kilobase of transcript per million reads mapped (RPKM) values. The *x*-axis represents the days post anthesis (DPA). Data are presented as mean ± standard deviation of three biological replicates for *O. meridionalis* and *O. sativa* and four biological replicates for *O. rufipogon* type taxa (except for 25 DPA with three replicates). * symbol above the line between two DPA represents significant differential gene expression and (*) with a circle represents significant differential gene expression between 5 DPA and 25 DPA.

**Figure 5 ijms-23-13351-f005:**
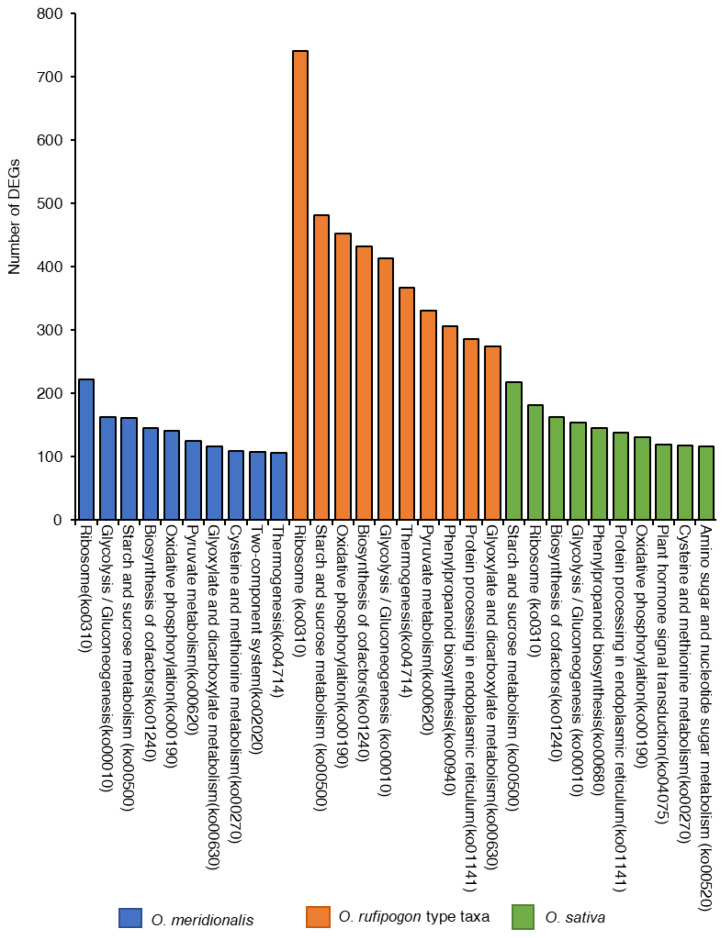
Top ten Kyoto Encyclopedia of Genes and Genomics (KEGG) pathways represented by the highest number of differentially expressed genes (DEGs) in *O. meridionalis*, *O. rufipogon* type taxa, and *O. sativa*.

## Data Availability

The datasets presented in this study can be found in online repositories. The names of the repository/repositories and accession number(s) can be found below: https://www.ncbi.nlm.nih.gov/genbank/, BioProject number PRJNA819483.

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
