# Peer review of "Gene Expression in the Developing Seed of Wild and Domesticated Rice"

_ijms, 2022, doi:10.3390/ijms232113351_

Round 1
Reviewer 1 Report
1. The manuscript used only one domesticated rice and two Australian wild rice taxa, are the results representative?
2. In addition to sequencing analysis, what is innovative about the article?
3. Lines 37-41 require references, many other sections do not have references.
4. The discussion section does not cite other similar literature to discuss, more like a report of experimental results. Major revisions are required.
Author Response
- The manuscript used only one domesticated rice and two Australian wild rice taxa, are the results representative?
Response: The results are representative as the study determines the timing of gene expression in two novel Australian wild rice taxa in comparison to one domesticated rice. These wild taxa represent a novel gene pool. It is likely that no genetic exchange has occurred between these wild taxa and domesticated rice, unlike Asian wild rice that has grown together with domesticated rice for many thousands of years. Therefore, these wild rice taxa are more reliable representatives of wild rice and provide a unique opportunity to gain a better understanding on how genes are differentially expressed in seed development stages in wild rice as compared to domesticated rice.
- In addition to sequencing analysis, what is innovative about the article?
Response: Very little research has been conducted on expression of genes related to domestication, starch biosynthesis and metabolism and seed storage protein. No other study has tracked changes in the transcriptome of developing rice seeds(grains) through development. This is also the very first time this has been analyzed in wild rice. We also report some transcripts from later stages of seed of the two wild rice taxa and domesticated rice that are novel (not reported previously).
- Lines 37-41 require references, many other sections do not have references.
Response: Added as suggested.
- The discussion section does not cite other similar literature to discuss, more like a report of experimental results. Major revisions are required.
Response: There is no other research on transcriptome analysis in seeds of wild rice and domesticated rice. We have cited all the related literature that is available.
Reviewer 2 Report
1. The abstract need to be improved as some of the lines mentioning DEGs are repetitive. please make it more specific.
2. The study has identified several novel genes,regulatory and metabolic pathways in developing seeds . However no discussion is provided regarding which gene/s could be appropriate candidate for modification/editing and utilization in domestication of the wild species.
3. The discussion need to be improved highlighting the above points. Also the differences between the cultivated and wild as revealed from this study should be mentioned with their importance in domestication and re-domestication.
3. The discussion just mentions what was observed in expression profiling of known and unknown genes and nothing further. Kindly elaborate the importance of your results in terms of new insights in plant physiology and development . how the work may contribute in future towards increasing genetic base and diversity of rice.
4. A concrete take home message at the end of abstract is required.
Author Response
- The abstract need to be improved as some of the lines mentioning DEGs are repetitive. please make it more specific.
Response: The Abstract has now been rewritten to limit the repetitiveness of DEGs.
- The study has identified several novel genes, regulatory and metabolic pathways in developing seeds . However no discussion is provided regarding which gene/s could be appropriate candidate for modification/editing and utilization in domestication of the wild species.
Response: The focus of this study is to determine the timing of gene expression in wild rice and domesticated rice and to characterize the transcriptome of the developing rice seed for the first time. It is not focused specifically on finding genes that could be used in domestication of wild rice. As discussed, this study provides new insights into the processes involved in seed development and also the impact of domestication. This is contribution made in response to a request for a contribution at no processing charge to a Special Issue: Rice Molecular Breeding and Genetics Guest Editors: Dr. Deyong Ren. The topic seems open to any contribution on rice genetics and is not limited to domestication.
- The discussion need to be improved highlighting the above points. Also the differences between the cultivated and wild as revealed from this study should be mentioned with their importance in domestication and re-domestication.
Response: As the objective of this research is not to determine candidate genes required for domestication or re-domestication, we didn’t discuss it in the manuscript. The type of study suggested would require a different approach.
- The discussion just mentions what was observed in expression profiling of known and unknown genes and nothing further. Kindly elaborate the importance of your results in terms of new insights in plant physiology and development . how the work may contribute in future towards increasing genetic base and diversity of rice.
Response: The results provide insight into the timing of the gene expression of genes related to domestication traits, starch biosynthesis and metabolism and seed storage proteins. We have now added more discussion of the potential to contribute to the capture of genetic diversity in rice breeding.
- A concrete take home message at the end of abstract is required.
Response: The last line of the abstract now summarises the core finding of the study.
Round 2
Reviewer 1 Report
Accept in present form.
Reviewer 2 Report
Accept for publication.